# CONDITIONAL SET GENERATION USING SEQ2SEQ MODELS

## ABSTRACT

Conditional set generation learns a mapping from an input sequence of tokens to a set. Several popular natural language processing (NLP) tasks, such as entity typing and dialogue emotion tagging, are instances of set generation. Sequence-to-sequence models are a popular choice to model set generation but this typical approach of treating a set as a sequence does not fully leverage its key properties, namely order-invariance and cardinality. We propose a novel data augmentation approach that recovers informative orders for labels using their dependence information. Further, we jointly model the set cardinality and output by listing the set size as the first element and taking advantage of the autoregressive factorization used by SEQ2SEQ models. Our experiments in simulated settings and on three diverse NLP datasets show that our method improves over strong SEQ2SEQ baselines by about 9% on absolute F1 score. We will release all code and data upon acceptance.

## 1 INTRODUCTION

Conditional set generation is the task of modeling the distribution of an output set given an input sequence of tokens (Kosiorek et al., 2020). Several natural language processing (NLP) tasks are instances of set generation, including open-entity typing (Choi et al., 2018; Dai et al., 2021) and fine-grained emotion classification (Demszky et al., 2020). The recent successes of pretraining-finetuning paradigm have encouraged a formulation of set generation as a sequence-to-sequence generation task (Vinyals et al., 2016; Yang et al., 2018; Ju et al., 2020).

In this paper, we argue that modeling set generation as a vanilla SEQ2SEQ generation task is sub-optimal as the SEQ2SEQ formulations do not explicitly account for two key properties of a set output: *order-invariance* and *cardinality*. Forgoing order-invariance, vanilla SEQ2SEQ generation modeling treats a set as a sequence, and thus assumes an arbitrary order between the elements it outputs. Similarly, the cardinality of sets is ignored, as the number of elements to be generated is typically not explicitly modeled. Although prior work has highlighted the importance of modeling the order-invariant nature of both set inputs (Zaheer et al., 2017) and outputs (Vinyals et al., 2016; Rezatofighi et al., 2018), the question of effectively modeling set output using SEQ2SEQ models still remains an open challenge.[1]

Our method addresses the challenges above by taking advantage of the auto-regressive factorization used by SEQ2SEQ models and (i) imposing an *informative* order over the label space, and (ii) explicity modeling *cardinality*. First, the label sets are converted to sequences using informative orders by grouping labels and leveraging their dependency structure. A natural way to model this is to search exhaustively for the best label orders. To efficiently search for such informative orders over a combinatorial space, our method imposes a partial order graph over the labels, where the nodes are the labels and the edges denote the conditional dependence relations. We then generate the training data with a fixed input and orders over the label set that are sampled by performing topological traversals over the graph. Labels that are not constrained by dependency relations are augmented in different positions in each sample, reinforcing the order-invariance. We then create an augmented training dataset, where each input instance is paired with various valid label sequences sampled from the dependency graph. Next, we jointly model a set with its cardinality by simply appending the size of the set as the first element in the sequence.

---

[1] Our work focuses on settings where the input is a sequence, and the output is a set.

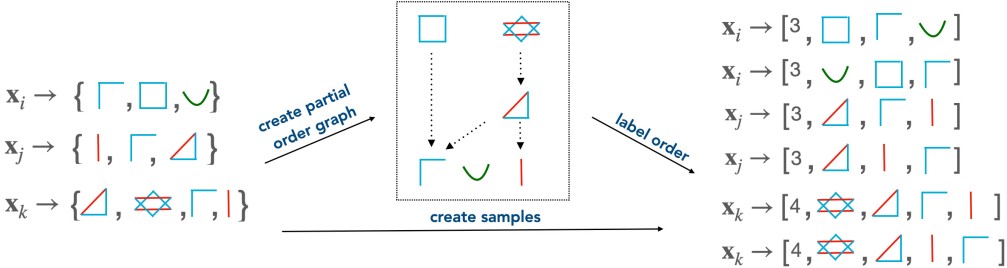

Figure 1: The figure illustrates a sample task where given an input $x$, the output is a set of shapes (e.g., triangle, half-square, line). The partial order graph (middle) arranges the label space such that specific labels (triangle) come before more general labels (line). Listing the specific labels first gives the model more clues about the rest of the set, leading to more informative sequences. The size of each set is also added as the first element for joint modeling of output with size.

Figure 1 illustrates the key intuitions behind our method using sample task where given an input $x$, the output is a set of shapes and their constituents ($\mathbb{Y}$). To see why certain orders might be more meaningful, consider a case where the output is a triangle consisting of a half-square and a line. After first generating triangle as a shape, the model can generate a half-square with certainty (a triangle will always contain a half-square). In contrast, the reverse order (generating half-square first) still leaves room for two possible shapes: square and triangle. The order [triangle, half-square] is thus more informative than [half-square, triangle]. The cardinality of a set can also be helpful. In our example, a triangle is composed of two shapes, and a star with three. A model that first predicts the number of shapes to generate can be more precise in its output and avoid over-generation, a major challenge with language generation models (Welleck et al., 2019; Fu et al., 2021).

Empirically, we establish the utility and soundness of our approach by showing gains on three real-world NLP datasets ($\sim$10% in $F$–scores). This result is significant - we effectively show that simple techniques such as augmenting cardinality and automated data augmentation approaches can substantially improve sequence to set generation tasks without any additional annotation overhead or architecture changes. We also provide a theoretical grounding for our approach. Treating the order as a latent variable, we show that TSAMPLE serves as a better proposal distribution when viewed via a variational inference framework. Finally, we perform an in-depth analysis of the reasons behind the sensitivity of the SEQ2SEQ framework on order by experimenting with a simulated experiment that realistically mimics a conditional set generation setting.

**Our contributions** (i) we show an efficient way to model sequence-to-set prediction as an SEQ2SEQ task by jointly modeling the cardinality and proposing a novel TSAMPLE data augmentation approach to add informative sequences. (ii) we show theoretically and empirically that our approach is better suited for set generation tasks than existing approaches.

## 2 BACKGROUND AND RELATED WORK

**Notation** Our focus is the setting where we are given a corpus $\mathcal{D}$ of $\{(\boldsymbol{x}_t, \mathbb{Y}_t)\}_{t=1}^m$ where $\boldsymbol{x}_t$ is a sequence of tokens and $\mathbb{Y}_t = \{y_1, y_2, \ldots, y_k\}$ is a set. For example, in multi-label fine-grained sentiment classification, $\boldsymbol{x}_t$ is a paragraph, and $\mathbb{Y}_t$ is a set of sentiments expressed by the paragraph. We use $y_i$ to denote an output symbol, $[y_i, y_j, y_k]$ to denote an ordered sequence of symbols and $\{y_i, y_j, y_k\}$ to denote a set.

### 2.1 SET GENERATION USING SEQ2SEQ MODEL

**Task** Given a corpus $\{(\boldsymbol{x}_t, \mathbb{Y}_t)\}_{t=1}^m$, the task of conditional set generation is to efficiently estimate $p(\mathbb{Y}_t \mid \boldsymbol{x}_t)$.

In this work, we adopt SEQ2SEQ models for the task. SEQ2SEQ models factorize $p(\mathbb{Y}_t \mid \boldsymbol{x}_t)$ in an autoregressive (AR) fashion using the chain rule:

$$p(\mathbb{Y}_t \mid \boldsymbol{x}_t) = p(\mathrm{y}_1, \mathrm{y}_2, \ldots, \mathrm{y}_k \mid \boldsymbol{x}_t)$$

$$= p(\mathrm{y}_1 \mid \boldsymbol{x}_t) \prod_{j=2}^{k} p(\mathrm{y}_j \mid \boldsymbol{x}_i, \mathrm{y}_1 \ldots \mathrm{y}_{j-1}) \qquad (1)$$

where we have used the order $\mathbb{Y}_t = [\mathrm{y}_1, \mathrm{y}_2, \ldots, \mathrm{y}_k]$ to factorize the joint distribution using chain rule. In theory, any of the $k!$ orders can be used to factorize the same joint distribution. In practice, however, the choice of order is important. For instance, Vinyals et al. (2016) show that output order affects language modeling performance when using LSTM based SEQ2SEQ models for set generation.

Consider an example $(\boldsymbol{x}_t, \mathbb{Y}_t = \{\mathrm{y}_1, \mathrm{y}_2\})$ pair. By chain rule, we have the following equivalent factorizations of this sequence: $p(\mathbb{Y}_t \mid \boldsymbol{x}_t) = p(\mathrm{y}_1 \mid \boldsymbol{x})p(\mathrm{y}_2 \mid \boldsymbol{x}, \mathrm{y}_1) = p(\mathrm{y}_2 \mid \boldsymbol{x})p(\mathrm{y}_1 \mid \boldsymbol{x}, \mathrm{y}_2)$. However, order-invariance is only guaranteed with *true* conditional probabilities, whereas the conditional probabilities used to factorize a sequence are *estimated* by a model from a corpus. Thus, depending on the order, the sequence factorizes as either $\hat{p}(\mathrm{y}_1 \mid \boldsymbol{x})\hat{p}(\mathrm{y}_2 \mid \boldsymbol{x}, \mathrm{y}_1)$ or $\hat{p}(\mathrm{y}_2 \mid \boldsymbol{x})\hat{p}(\mathrm{y}_1 \mid \boldsymbol{x}, \mathrm{y}_2)$, which are not necessarily equivalent. Further, one of the two factorizations might closely approximate the true distribution, thus being a better choice.

## 2.2 EXISTING TECHNIQUES FOR SET GENERATION

Set generation for computer vision problems has received considerable attention. Specifically, Rezatofighi et al. (2018; 2020) investigate set outputs for vision tasks. Their learning procedure involves jointly learning the order and the cardinality of the set. However, their method relies on searching through a combinatorial space of permutations.

Zhang et al. (2019a) propose deep set prediction networks (DSPN), using an auto-encoder framework with a set encoder for conditional generation of digits and image tags with a fixed maximum number of elements. Kosiorek et al. (2020) extend DSPN by additionally modeling the cardinality of the output using an MLP. Finally, Zhang et al. (2020) explore the usage of energy-based models for set prediction. Their learning and inference procedure relies on drawing samples from the set distribution, which is prohibitively expensive for extremely high-dimensional spaces like text. Other examples include works such as Salvador et al. (2019), who aim to extract set of ingredients from food images.

Our approach differs from their work in several important ways: i) instead of performing an exhaustive search over the sample space, we add informative order over labels in the input as a data augmentation step, ii) we model cardinality simply by listing the set size as the first element of the sequence, and thus jointly learn both it with the set output, and iii) Image classification and tagging typically involves a small, independent number of tags. In contrast, NLP tasks have richer and larger label space. Our method is more suitable for such tasks as it does not rely on exhaustive search over label space and leverages label dependencies.

Chen et al. (2021) explored the generation of an optimal order for graph generation *given* the nodes. They observed that ordering nodes before inducing edges improves graph generation. However, in our case, since the labels themselves are being generated, conditioning on the labels to create the optimal order is not possible for non-trivial setups.

**Non-SEQ2SEQ set generation** These include using deep reinforcement learning for multi-label classification (Yang et al., 2019) and combinatorial problems such as Sudoku (Nandwani et al., 2020), and pointer networks (Ye et al., 2021) for extracting and generating keyphrases. Unlike these works, our focus is on methods that can optimally adapt existing SEQ2SEQ models for set generation, without doing using external knowledge (Wang et al., 2020; Zhang et al., 2019b). Since our approach does not involve directly changing the model parameters or training procedure, we can leverage the advantages of the pretraining-finetuning paradigm and large-scale language models, which have shown immense promise in several NLP tasks.

**Connection with Janossy pooling** Murphy et al. (2019) generalize deep sets by proposing to encode a set of $N$ elements by pooling permutations of $P(N, k)$ tuples. With $k = N$, their method

is the same as pooling all $N!$ sequences, and with $k = 1$, it reduces to deep sets. Our approach shares the spirit of tractable searching over $N!$ with Janossy pooling. However, instead of iterating over all possible 2-tuples, our method imposes pairwise constraints on the order of the elements.

### 2.3 Modeling set input

A number of techniques have been proposed for encoding set-shaped inputs (Santoro et al., 2017; Zaheer et al., 2017; Lee et al., 2019; Murphy et al., 2019; Huang et al., 2020; Kim et al., 2021). Specifically, Zaheer et al. (2017) propose deep sets, wherein they show that pooling the representations of individual set elements and feeding the resulting features to a non-linear network is a principled way of representing sets. Lee et al. (2019) present permutation-invariant attention to encode shapes and images using a modified version of attention (Vaswani et al., 2017). We note that our work focuses on settings where the input is a sequence, and the output is a set.

## 3 Method

In this section, we present TSAMPLE, a novel method that tractably creates informative orders over sets. We also present our approach of jointly modeling cardinality and set output.

### 3.1 Adding informative orders for set output

As discussed in Section 2, SEQ2SEQ formulation requires the output to be in a sequence. Prior work (Vinyals et al., 2016; Rezatofighi et al., 2018; Chen et al., 2021) has noted that adding orders that have the highest conditional likelihood given the input is an optimal choice. Unlike these methods, we create training data using orders sampled from TSAMPLE, thus completely sidestepping exhaustive searching during training.

Our core insight is that knowing the optimal order between pairs of symbols in the output drastically reduces the possible number of permutations. We thus impose pairwise order constraints for a subset of labels. Specifically, given an output set $\mathbb{Y}_t = y_1, y_2, \ldots, y_k$, if $y_i, y_j$ are independent, they can be added in an arbitrary order. Otherwise, an order constraint is added to the order between $y_i, y_j$.

**Learning pairwise constraints** We estimate the dependence between elements $y_i, y_j$ using pointwise mutual information: $\mathtt{pmi}(y_i, y_j) = \log p(y_i, y_j)/p(y_i)p(y_j)$. Here, $\mathtt{pmi}(y_i, y_j) > 0$ indicates that the labels $y_i, y_j$ co-occur more than would be expected under the conditions of independence (Wettler & Rapp, 1993). We use $\mathtt{pmi}(y_i, y_j) > \alpha$ to filter our such pairs of dependent pairs, and perform another check to determine if the order between them should be fixed. For each dependent pair $y_i, y_j$, the order is constrained to be $[y_i, y_j]$ if $\log p(y_j \mid y_i) - \log p(y_i \mid y_j) > \beta$ ($y_j$ should come after $y_i$), and $[y_j, y_i]$ otherwise. Intuitively, $\log p(y_j \mid y_i) - \log p(y_i \mid y_j) > \beta$ implies that knowledge that a set contains $y_i$, increases the probability of $y_j$ being present. Thus, fixing the order to $[y_i, y_j]$ will be more efficient for generating a set with $\{y_i, y_j\}$.

**Generating samples** To systematically create permutations that satisfy these constraints, we construct a topological graph $G_t$ where each node is a label $\boldsymbol{y}_i \in \mathbb{Y}_t$, and the edges are determined using the $\mathtt{pmi}$ and the conditional probabilities as outlined above (Algorithm 1). The required permutations can then simply be generated as topological traversals $G_t$ (Figure 2). To generate diverse samples, we begin the traversal from a different starting node. We call this method TSAMPLE. Later, we show that TSAMPLE can be interpreted as a proposal distribution in variational inference framework, which distributes the mass uniformly over informative orders constrained by the graph.

**Do pairwise constraints hold for longer sequences?** While TSAMPLE uses pairwise (and not higher-order) constraints for ordering variables, we note that the pairwise checks remain relevant with extra variables. First, dependence between pair of variables is retained in joint distributions involving more variables ($y_i \not\perp\!\!\!\perp y_j \implies y_i \not\perp\!\!\!\perp y_j, \boldsymbol{y}_k$) for some $\boldsymbol{y}_k \in \mathbb{Y}$ (Appendix A.1). Further, if $y_i, y_j \perp\!\!\!\perp \boldsymbol{y}_k$, then it can be shown that $p(y_i \mid y_j) > p(y_j \mid y_i) \implies p(y_i \mid y_j, \boldsymbol{y}_k) > p(y_j \mid y_i, \boldsymbol{y}_k)$ (Appendix A.2). The first property shows that the pairwise dependencies hold in the presence of other elements of the set. The second property shows that an informative order continues

to be informative when additional independent symbols are added to it. Thus, our criterion of using pairwise dependencies between the elements of a set is still effective. Finally, we note that using higher-order dependencies might be suboptimal for practical reasons: higher-order dependencies (or including $\boldsymbol{x}_t$) might not be accurately discovered due to sparsity, and thus causing spurious orders.

---

**Algorithm 1** Generating permutations for $\mathbb{Y}_t$

**Input**: Set $\mathbb{Y}_t$, number of permutations $n$
**Parameter**: $\alpha, \beta$
**Output**: $n$ topological sorts over $G_t(V, E)$

1: Let $V = \mathbb{Y}_t, E = \emptyset$.
2: **for** $\mathrm{y}_i, \mathrm{y}_j \in \mathbb{Y}_t$ **do**
3:    **if** $pmi(\mathrm{y}_i, \mathrm{y}_j) > \alpha$ and $\log p(\mathrm{y}_i \mid \mathrm{y}_j) - \log p(\mathrm{y}_j \mid \mathrm{y}_i) > \beta$ **then**
4:       $E = E \cup \mathrm{y}_j \to \mathrm{y}_i$
5:    **end if**
6: **end for**
7: **return** $\texttt{topo\_sort}(G_t(V, E), n)$

---

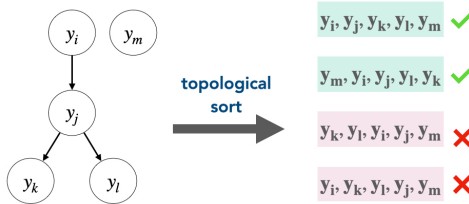

Figure 2: Our method first builds a graph $G_t$ over the set $\mathbb{Y}_t$, and then samples orders from $G_t$ using topological sort ($\texttt{topo\_sort}$). The topological sorting rejects samples that do not follow the conditional probability constraints.

**Complexity analysis** Let $\mathbb{Y}$ be the label space (i.e., set of all possible labels), $(\boldsymbol{x}_t, \mathbb{Y}_t)$ be a particular training example, $N$ be the size of the training set, and $c$ be the maximum number of elements for any set $\mathbb{Y}_t$ in the input. Our method requires three steps: i) iterating over the training data to learn conditional probabilities and pmi, and ii) given a $\mathbb{Y}_t$, building the topo-graph $G_t$ (Algorithm 1), and iii) traversing $G_t$ to create samples for $(\boldsymbol{x}_t, \mathbb{Y}_t)$.

The time complexity of the first operation is $\mathcal{O}(Nc^2)$: for each element of the training set, the pairwise count for each pair $\mathrm{y}_i, \mathrm{y}_j$ and unigram count for each $\mathrm{y}_i$ is calculated. The pairwise counts can be used for calculating joint probabilities. In principle, we need $\mathcal{O}(|\mathbb{Y}|^2)$ space for storing the joint probabilities, but only a small fraction of the possible combinations appear together in practice.

Given a set $\mathbb{Y}_t$, the graph $G_t$ is created in $\mathcal{O}(c^2)$ time. Then, generating $k$ samples from $G_t$ requires a topological sort, for $\mathcal{O}(kc)$ (or $\mathcal{O}(c)$ per traversal). For training data of size $N$, the total time complexity is $\mathcal{O}(Nck)$.

The entire process (building the joint counts and creating graphs and samples) takes less than five minutes for all datasets for our experiments (on an 80-core Intel Xeon Gold 6230 CPU) .

**Why should augmenting with permutations help?** We show that our method of augmenting permutations to the training data can be interpreted as an instance of variational inference with the order as a latent variable, and TSAMPLE as an instance of a richer proposal distribution. Let $\pi_j$ be the $j^{th}$ order over $\mathbb{Y}_t$ (out of $|\mathbb{Y}_t|!$ possible orders $\Pi$), and $\pi_j(\mathbb{Y}_t)$ be the sequence of elements in $\mathbb{Y}_t$ arranged with order $\pi_j$. Treating $\pi$ as a latent random variable, the output distribution can then be recovered by marginalizing over $\Pi$: $\log p_\theta(\mathbb{Y}_t \mid \boldsymbol{x}_t) = \log \sum_{\pi_z \in \Pi} p_\theta(\pi_z(\mathbb{Y}_t) \mid \boldsymbol{x}_t)$, $\Pi$: $\log p_\theta(\mathbb{Y}_t \mid \boldsymbol{x}_t) = \log \sum_{\pi_z \in \Pi} p_\theta(\mathbb{Y}_t, \pi_z \mid \boldsymbol{x}_t)$ where $p_\theta$ is the SEQ2SEQ conditional generation model. While summing over $\Pi$ is intractable, standard techniques from the variational inference framework allow us to write a lower bound (ELBO) on the actual likelihood:

$$\log p_\theta(\mathbb{Y}_t \mid \boldsymbol{x}_t) = \log \sum_{\pi_{\boldsymbol{z}} \in \Pi} p_\theta(\pi_z(\mathbb{Y}_t) \mid \boldsymbol{x}_t) \geq \underbrace{\mathbb{E}_{q_\phi(\pi_{\boldsymbol{z}})}\left[\frac{\log p_\theta(\pi_{\boldsymbol{z}}(\mathbb{Y}_t) \mid \boldsymbol{x}_t)}{q_\phi(\pi_{\boldsymbol{z}})}\right]}_{\text{ELBO}} = \mathcal{L}(\theta, \phi)$$

In practice, the optimization procedure draws $k$ samples from the proposal distribution $q$ to optimize a weighted ELBO (Burda et al., 2016; Domke & Sheldon, 2018). Crucially, $q$ can be fixed (e.g., to uniform distribution over the orders), and in such cases only $\theta$ are learned (Appendix C).

TSAMPLE can thus be seen as a particular proposal distribution that assigns all the weights to the topological ordering over the label dependence graphs. We also experiment with sampling from a

uniform distribution over the samples (referred to as UNIFORM experiments in our baseline setup). We note that the idea of using an informative proposal distribution over space of structures to do variational inference has also been used in the context of grammar induction (Dyer et al., 2016) and graph generation (Jin et al., 2018; Chen et al., 2021). Our formulation is closest in spirit to Chen et al. (2021). However, in their graph generation setting, the set of nodes to be ordered is already given. In contrast, we infer the order and the set elements jointly from the input.

### 3.2 MODELING CARDINALITY

Let $m = |\mathbb{Y}_t|$ be the cardinality of $\mathbb{Y}_t$ (or the number of elements in $\mathbb{Y}_t$). Our goal is to jointly estimate $m$ and $\mathbb{Y}_t$ (i.e., $p(m, \mathbb{Y}_t \mid \boldsymbol{x}_t)$). Additionally, we want the model to use the cardinality information for generating $\mathbb{Y}_t$. To this end, we simply add the order information at the beginning of the sequence. That is, we convert a sample $(\boldsymbol{x}_t, \mathbb{Y}_t)$ to $(\boldsymbol{x}_t, [|\mathbb{Y}_t|, \pi(\mathbb{Y}_t)])$, and then train our SEQ2SEQ model as usual from $\boldsymbol{x} \rightarrow [|\mathbb{Y}_t|, \pi(\mathbb{Y}_t)]$. As SEQ2SEQ models use autoregressive factorization, listing the order information first ensures that the sequence factorizes as $p([|\mathbb{Y}_t|, \pi(\mathbb{Y}_t)] \mid \boldsymbol{x}_t) = p(|\mathbb{Y}_t| \mid \boldsymbol{x}_t)p(\pi(\mathbb{Y}_t) \mid |\mathbb{Y}_t|, \boldsymbol{x}_t)$. Thus, the generation of $\mathbb{Y}_t$ is conditioned on both the input and the cardinality as desired (note the $p(\pi(\mathbb{Y}_t) \mid |\mathbb{Y}_t|, \boldsymbol{x}_t)$ term).

**Why should cardinality help?** Unlike models like deep sets (Zhang et al., 2019a), SEQ2SEQ models are not restricted by the number of elements generated in the output. However, the information about the number of elements to be generated has two potential benefits: i) it can help avoid over-generation (Welleck et al., 2019; Fu et al., 2021), and ii) unlike free-form text output, the distribution of the set output size ($p(|\mathbb{Y}_t| \mid \boldsymbol{x}_t)$) might benefit the model to adhere to the set size constraint. Thus, information on the predicted size can be beneficial for the model to predict the elements to be generated. In the following section, we extensively test our proposed method via a simulated setting and empirical analysis on diverse real-world datasets.

## 4 EXPERIMENTS

### 4.1 SIMULATION

We design a simulation to investigate the effects of output order and cardinality on conditional set generation, following prior work that has found simulation to be an effective for studying properties of deep neural networks (Vinyals et al., 2016; Khandelwal et al., 2018).

**Data generation** We use a graphical model (Figure 3) to generate conditionally dependent pairs $(\boldsymbol{x}, \mathbb{Y})$, with different levels of interdependencies among the labels in $\mathbb{Y}$. Let $\mathbb{Y} = \{y_1, y_2, \ldots, y_N\}$ be the label space (i.e., label space ). We sample

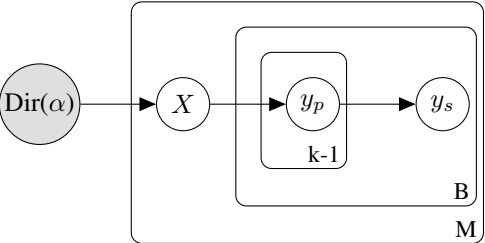

Figure 3: Generative process for simulation.

a dataset of the form $\{(\boldsymbol{x}, \boldsymbol{y})\}_{i=1}^{m}$. $\boldsymbol{x}$ is an $N$ dimensional multinomial sampled from a dirichlet parameterized by $\alpha$. The output set $\boldsymbol{y} = \{y_1, y_2, \ldots, y_{Bk}\}$ is created in $B$ *blocks*, each block of size $k$ and $y_i \in \mathbb{Y}$. A block is created by first sampling $k - 1$ labels ($\boldsymbol{y}_p$) independently from Multinomial($\boldsymbol{x}$). The $k^{th}$ label ($y_s$) is sampled from either a uniform distribution with a probability $= \epsilon$ or is deterministically determined from the preceding $k - 1$ labels. For block size of 1 ($k = 1$), the output is simply a set of size $B$ sampled from $\boldsymbol{x}$ where all the labels are independent. Similarly, $k = 2$ simulates a situation with a high degree of dependence: each block is of size 2, with $\boldsymbol{y}_p$ sampled independently from the input, and the $y_s$ determined deterministically from $\boldsymbol{y}_p$. . Gradually increasing the block size increases the number of independent elements.

### 4.1.1 SIMULATION RESULTS

We use the architecture of BART-base (Lewis et al., 2020) without pre-training for all simulations[2].

---

[2] All the simulations were repeated using three different random seeds, and we report the averages.

**TSAMPLE leads to higher set overlap and helps across all sampling types**: To test our method against UNIFORM, we use *perplexity* and *jaccard coefficient*. Jaccard coefficient captures the ability of the model to generate more informative sequences, whereas perplexity measures model's sensitivity to order. We gradually augment the training data with orders sampled from a uniform distribution over orders (UNIFORM) and TSAMPLE, and evaluate the learning and the final set overlap using training perplexity and Jaccard score, respectively. The results show that augmentations done using TSAMPLE help the model converge faster, and to a lower perplexity (Figure 4 left). TSAMPLE also consistently outperforms UNIFORM across block sizes (Figure 4 right). We observe that the efficacy of TSAMPLE reduces with increasing block size. This can be understood by noting that as the number of independent elements increase, the effect of order on the joint distribution diminishes (proof in Appendix A.3). Further, we found that TSAMPLE is not sensitive to the sampling type: across five different sampling types, including nucleus (Holtzman et al., 2020) and greedy sampling, augmenting with TSAMPLE permutations yields significant gains (Table 5 in Appendix E).

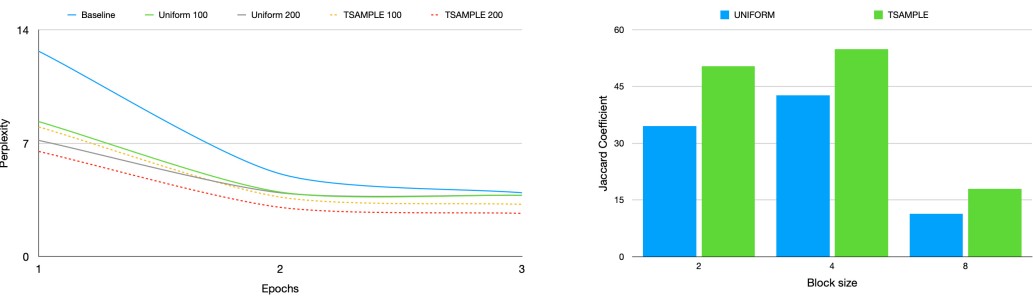

Figure 4: Effect of TSAMPLE on perplexity (left) and set overlap (right).

**SEQ2SEQ models can learn cardinality and use it for better decoding** : We created sample data from Figure 3 where the length of the output is determined by sum of the inputs $X$. We experimented with and without including cardinality as the first element. We found that training with cardinality increases step overlap by over 15%, from 40.54 to 46.13. Further, the version with cardinality accurately generated sets which had the same length as

|  | avg/min/max labels per sample | unique labels | train/test/dev samples per split |
|---|---|---|---|
| GO-EMO | 3.03/3/5 | 28 | 0.6k/0.1k/0.1k |
| OPENENT | 5.4/2/18 | 2519 | 2k/2k/2k |
| REUTERS | 2.52/2/11 | 90 | 0.9k/0.4k/0.3k |

Table 1: Dataset statistics.

the target 70.64% of the times, as opposed to 27.45% for the version without cardinality. A number of other findings, including conditions where order matters the most, effect of randomness and independence on our task are included in Appendix E.

## 4.2 REAL-WORLD TASKS

To establish the efficacy of our approach in real-world data settings, we experiment with three different multi-label classification tasks (examples in Table 3):

- Go-Emotions classification (GO-EMO, Demszky et al. (2020)): Generating a set of emotions for an input paragraph.
- Open Entity Typing (OPENENT, Choi et al. (2018)): Assigning open types (free-form phrases) to the tagged entities in the input text. Here, the set of possible entity types is open, this task allows us to investigate our method in situations where the label space is not constrained.
- Reuters-21578 (REUTERS, Lewis (1997)): A collection of newswire documents from Reuters, where each article has to be labeled with a set of economic subjects mentioned in it.

We treat all the problems as open-ended generation problems, and do not use any specialized preprocessing. For all the datasets, we filter out samples with a single label. For each training sample, we create $n$ permutations over TSAMPLE to create the training data.

**Baselines**  We experiment with the following four baselines (Table 2):

- BART-MULTI-LABEL: A multi-label classifier where the input is encoded using bart-base, and used to make independent (pointwise) predictions the output labels. This baseline represents the standard method for doing multi-label classification (e.g., Demszky et al. (2020)). During inference, we take top $k = [1, 3, 5, 10, 50]$ labels as the true labels, and report the average (Table 9 shows experiments with bert-base-uncased).
- SET SEARCH: each training sample $(\boldsymbol{x}, \{y_1, y_2, \ldots, y_k\})$ is converted into $k$ different training examples $\{(\boldsymbol{x}, y_i)\}_{i=1}^{k}$. During inference, unique elements generated by beam search are returned as the set output. The size of the beam is set to the maximum possible set size in the training data (Table 1). This is a popular approach for one-to-many generation tasks (Hwang et al., 2021).
- SEQ2SEQ: set elements are listed in a random order, and each sample is repeated $n$ times.
- UNIFORM: $n$ permutations are uniformly sampled from the possible permutations of labels.

**Model**  We use BART-base (Lewis et al., 2020) with pre-trained weights for all the tasks. We use $n = 2$ for TSAMPLE and UNIFORM. For all the results, we use three epochs and the same number of training samples. This controls for models trained with augmented data improving only because of factors such as longer training time. All the experiments were repeated for three different random seeds, and we report the averages. We conduct a one-tailed proportion of samples test (Johnson et al., 2000) to compare the best model with SEQ2SEQ (we do not use SET SEARCH for calculating significance) and underscore all results that are significant with $p < 0.0005$. For Algorithm 1, we experiment with $\alpha = \{0.5, 1, 1.5\}$ and $\beta = \{\log_2(2), \log_2(3), \log_2(4)\}$, and use the implementation of topological sort provided by networkx (Hagberg et al., 2008) and ignore cycles. We found from our experiments that hyperparameter tuning over $\alpha, \beta$ did not affect the results in any significant way. For all the experiments reported, we use $\alpha = 1$ and $\beta = \log_2(3)$. We use a single GeForce RTX 2080 Ti for all our experiments. Additional hyperparameter details in Appendix D.

**Results**  Table 2 summarizes the empirical results on the tasks. We report macro precision, recall, and $F$-measure on individual datasets. We observe that across all the datasets, incorporating cardinality and using TSAMPLE improves the performance significantly. When used with baseline approaches across all the datasets, modeling cardinality as part of the output provides significant performance gains. To complement, our TSAMPLE further improves the performance across datasets. More specifically, we observe that both precision and recall improves, showing the overall efficacy of our approach. TSAMPLE improves over UNIFORM and SEQ2SEQ by about 1% absolute $F$-score on average. Modeling cardinality provides a consistent performance gain of about 6% for SEQ2SEQ, 6% for UNIFORM, and 8% $F$-score for TSAMPLE. Overall, we achieve a net gain of 9% absolute $F$-score by incorporating both informative orders and cardinality.

In further analysis, we observed that the comparatively lower performance of SET SEARCH baseline is due to two specific reasons - repeated generation of the same set of terms (e.g., *person, business* for OPENENT) and generating elements not present in the test set. We also note that UNIFORM does not improve over SEQ2SEQ consistently (both with and without CARD), showing that merely augmenting with random permutations does not help.

### 4.3   ANALYSIS

To understand the nature of the label dependencies, we use qualitative examples from the datasets for an in-depth analysis. For this analysis, we selected a random subset of 100 samples from each of the datasets from the validation set.

**What kinds of permutations does TSAMPLE create?**  As discussed in Section 3.1, TSAMPLE encourages highly co-occuring pairs $(y_i, y_j)$ to be in the order $y_i, y_j$ if $p(y_j \mid y_i) > p(y_i \mid y_j)$. In our analysis, this dependency in the datasets shows that the orders exhibit a pattern where *specific* labels appear before the *generic* ones. For example, in case of entity typing, the more generic entity *event* is generated after the more specific entities *home game* and *match* Figure 4.3 (left).

**Increasing the number of permutations ($n$)**  We compare TSAMPLE and UNIFORM as $n$ increases from $n = 2$ to 10. Figure 4.3 (right) shows that both TSAMPLE and UNIFORM improve as $n$ is increased, with TSAMPLE outperforming UNIFORM across $n$.

| | GO-EMO | | | OPENENT | | | REUTERS | | |
|---|---|---|---|---|---|---|---|---|---|
| | $p$ | $r$ | $F$ | $p$ | $r$ | $F$ | $p$ | $r$ | $F$ |
| BART-MULTI-LABEL | 20.8 | 42.4 | 22.4 | 16.4 | 25.1 | 14.3 | 19.7 | 43.4 | 21.7 |
| SET SEARCH | 10.7 | 7.0 | 7.4 | 26.5 | 31.4 | 26.3 | 10.9 | 7.1 | 7.5 |
| SEQ2SEQ | 27.4 | 26.2 | 23.4 | 55.4 | 42.4 | 44.6 | 24.8 | 13.8 | 15.6 |
| UNIFORM | 32.5 | 19.9 | 22.7 | 62.6 | 41.7 | 46.9 | 26.7 | 12.7 | 15.2 |
| TSAMPLE | **36.7** | 19.8 | 23.3 | 60.0 | 44.5 | 48.0 | 26.5 | 12.8 | 15.8 |
| SEQ2SEQ +CARD | 33.0 | 28.3 | 26.8 | 62.5 | 44.7 | 50.5 | 34.1 | 21.8 | 24.3 |
| UNIFORM + CARD | 35.6 | 26.5 | 27.5 | **68.6** | 42.3 | 50.4 | 35.3 | 22.1 | 24.7 |
| TSAMPLE + CARD | 36.1 | **30.5** | **30.0** | 65.5 | **47.5** | 53.5 | 36.7 | 24.1 | 26.7 |

Table 2: Our main results: using permutations generated by TSAMPLE and adding cardinality gives the best overall performance in terms of macro precision, recall, and $F$-score. BART-MULTI-LABELis the standard multi-label classification approach. Statistically significant results are underscored. CARD stands for cardinality.

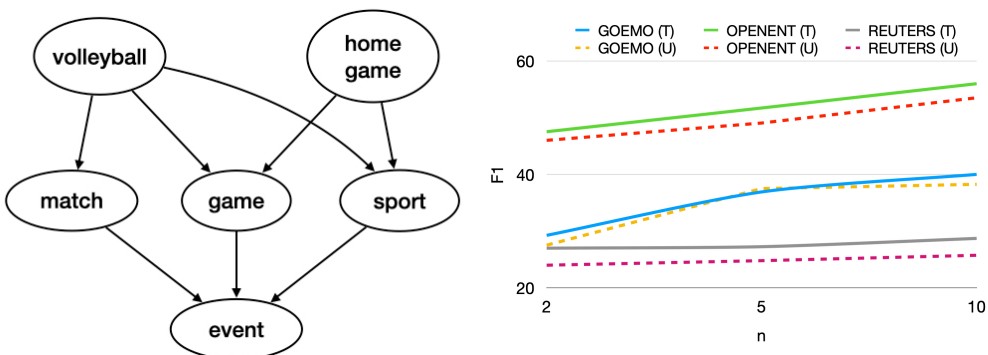

Figure 5: **Left:** label dependencies used by TSAMPLE for OPENENT: TSAMPLE puts specific entities (e.g., volleyball) before generic ones (e.g., event). **Right:** TSAMPLE ((T)) consistently outperforms UNIFORM ((U)) as $n$ is increased.

**Role of cardinality**   From the results in Table 2, we observe that cardinality is crucial to modeling set output. To study whether the models learn to condition on predicted set length, we compute an *agreement* score - defined as the % of times the predicted cardinality matches the number of elements generated by the model. We observe that the model effectively predicts the cardinality almost exactly in both GO-EMO and REUTERS datasets (average 95%). While the exact match agreement is low in OPENENT (35%), the model is within an error of $\pm 1$ in 93% of the cases.

**Reversing the order**   In order to check our hypothesis of whether only informative orders helping with set generation, we invert the label dependencies returned by TSAMPLE for all the datasets and train with the same model settings. Across all datasets, we observe that reversing the order leads to an average of 12% drop in $F$-score. The reversed order not only closes the gap between TSAMPLE and UNIFORM, but in many instances, the performance is slightly worse than UNIFORM.

## 5   CONCLUSION

We present a novel method for performing conditional set generation using SEQ2SEQ models that leverages both incorporating informative orders and adding cardinality information. Experiments in simulated settings and real-world datasets show that our method is more effective than strong baselines at set generation. We also present an in-depth analysis of our method along with the empirical results. In the future, we want to extend this work to explore better proposal distributions and to incorporate cardinality information in open-ended generation tasks like dialogue.

## ETHICS AND REPRODUCIBILITY STATEMENT

We take the following steps for reproducibility of our results:

1. All the experiments are performed for three different random seeds. In addition, we conduct a proportion of samples hypothesis test to establish the statistical significance of our results. We did not perform extensive hyperparameter tuning and used the same set of defaults for baselines and our proposed method.

2. For all data augmentation experiments, we match the number of training samples and epochs; all the models are trained for the same duration. This alleviates the concern that the models perform well with augmented data merely because of the longer training time.

3. We conduct a proportion of samples test for all the experiments conducted on real-world datasets and use a small $p = 0.0005$ to measure highly significant results, which are indicated with an underscore.

Our work aims to promote the usage of existing resources for as many use cases as possible. In particular, all our experiments are performed on the BASE-version of the model (BART) that can relatively lower parameter count to conserve resources and help lower our impact on climate change.

We propose a method to use existing pre-trained language models more efficiently for set generation. Our downstream datasets in this work do not contain any societally impactful or social themes. Hence, we do not anticipate any misuse as-is. To the best of our knowledge, we did not encounter any downstream tasks that can leverage our method for any negative impact. Despite that, it is certainly possible we might have missed something, and we are happy to engage anonymously with the reviewers, and the chairs and help address the concerns that may arise.

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

# A    PROOFS

Let $\mathbb{Y}$ be the output space, $\mathrm{y}_i, \mathrm{y}_j, \mathrm{y}_k \in \mathbb{Y}$, and $\boldsymbol{y}_k \in \mathbb{Y} - \mathrm{y}_i - \mathrm{y}_j$ be a subset of the symbols excluding $\mathrm{y}_i, \mathrm{y}_j$. We assume that all the distributions are non-negative (i.e., $p(\boldsymbol{y}) > 0, \forall \boldsymbol{y} \in \mathbb{Y}$)

**Lemma A.1** $\mathrm{y}_i \not\perp\!\!\!\perp \mathrm{y}_j \implies \mathrm{y}_i \not\perp\!\!\!\perp (\mathrm{y}_j \mathrm{y}_k)$

**Proof**    Let $\mathrm{y}_i \perp\!\!\!\perp (\mathrm{y}_j \mathrm{y}_k)$ by contradiction. Then:
$$p(\mathrm{y}_i, \mathrm{y}_j \mathrm{y}_k) = p(\mathrm{y}_i)p(\mathrm{y}_j \mathrm{y}_k) \tag{2}$$
Also,
$$\begin{aligned}
p(\mathrm{y}_i, \mathrm{y}_j) &= \sum_{\mathrm{y}_k \in \boldsymbol{Z}} p(\mathrm{y}_i, \mathrm{y}_j \mathrm{y}_k) \\
&= \sum_{\mathrm{y}_k \in \boldsymbol{Z}} p(\mathrm{y}_i)p(\mathrm{y}_j \mathrm{y}_k) && \text{(equation 2)} \\
&= p(\mathrm{y}_i) \sum_{\mathrm{y}_k \in \boldsymbol{Z}} p(\mathrm{y}_j \mathrm{y}_k) \\
&= p(\mathrm{y}_i)p(\mathrm{y}_j) \tag{3}
\end{aligned}$$
However, $\mathrm{y}_i \not\perp\!\!\!\perp \mathrm{y}$ thus $\mathrm{y}_i \not\perp\!\!\!\perp \mathrm{y} \implies \mathrm{y}_i \not\perp\!\!\!\perp (\mathrm{y}_j \mathrm{y}_k)$.

**Lemma A.2**
$$p(\mathrm{y}_i \mid \mathrm{y}_j) > p(\mathrm{y}_j \mid \mathrm{y}_i) \implies p(\mathrm{y}_i \mid \mathrm{y}_j, \boldsymbol{y}_k) > p(\mathrm{y}_j \mid \mathrm{y}_i, \boldsymbol{y}_k)$$
*if* $\mathrm{y}_i, \mathrm{y}_j \perp\!\!\!\perp \boldsymbol{y}_k$

**Proof**    We have:
$$\begin{aligned}
& p(\mathrm{y}_i \mid \mathrm{y}_j) > p(\mathrm{y}_j \mid \mathrm{y}_i) \\
\implies & p(\mathrm{y}_j) < p(\mathrm{y}_i) \tag{4}
\end{aligned}$$

$$\begin{aligned}
p(\mathrm{y}_j, \boldsymbol{y}_k) &= p(\boldsymbol{y}_k \mid \mathrm{y}_j)p(\mathrm{y}_j) \\
&< p(\boldsymbol{y}_k \mid \mathrm{y}_j)p(\mathrm{y}_i) && \text{(Equation 4)} \\
&= p(\boldsymbol{y}_k \mid \mathrm{y}_i)p(\mathrm{y}_i) && (\mathrm{y}_i, \mathrm{y}_j \perp\!\!\!\perp \boldsymbol{y}_k \implies p(\boldsymbol{y}_k \mid \mathrm{y}_j) = p(\boldsymbol{y}_k \mid \mathrm{y}_i) = p(\boldsymbol{y}_k)) \\
&= p(\mathrm{y}_i, \boldsymbol{y}_k) \tag{5}
\end{aligned}$$
Thus,
$$\begin{aligned}
p(\mathrm{y}_i \mid \mathrm{y}_j, \boldsymbol{y}_k) &= \frac{p(\mathrm{y}_i, \mathrm{y}_j, \boldsymbol{y}_k)}{p(\mathrm{y}_j, \boldsymbol{y}_k)} \\
&> \frac{p(\mathrm{y}_i, \mathrm{y}_j, \boldsymbol{y}_k)}{p(\mathrm{y}_i, \boldsymbol{y}_k)} \\
&= p(\mathrm{y}_j \mid \mathrm{y}_i, \boldsymbol{y}_k) \tag{6}
\end{aligned}$$

**Lemma A.3** *If* $\mathrm{y}_i \perp\!\!\!\perp \mathrm{y}_j \ \forall \mathrm{y}_i, \mathrm{y}_j \in \mathbb{Y}$*, the order is guaranteed to not affect learning.*

**Proof**    Let $\pi_j$ be the $j^{th}$ order over $\mathbb{Y}$ (out of $|\mathbb{Y}|!$ possible orders $\Pi$), and $\pi_j(\mathbb{Y})$ be the sequence of elements in $\mathbb{Y}$ arranged with $\pi_j$.
$$\begin{aligned}
p(\mathrm{y}_i \mid \mathrm{y}_j) &= p(\mathrm{y}_i) && (\mathrm{y}_i \perp\!\!\!\perp \mathrm{y}_j \ \forall \mathrm{y}_i, \mathrm{y}_j) \\
\implies p(\mathrm{y}_i, \mathrm{y}_j, \mathrm{y}_k) &= p(\mathrm{y}_i)p(\mathrm{y}_j \mid \mathrm{y}_i)p(\mathrm{y}_k \mid \mathrm{y}_i, \mathrm{y}_j) \\
&= p(\mathrm{y}_i)p(\mathrm{y}_j)p(\mathrm{y}_k) \\
\implies p(\pi_m(\mathrm{y}_i, \mathrm{y}_j, \mathrm{y}_k)) &= p(\pi_n(\mathrm{y}_i, \mathrm{y}_j, \mathrm{y}_k)) \ \forall \pi_m, \pi_m \in \Pi
\end{aligned}$$
In other words, when all elements are mutually independent, all possible joint factorizations will simply be a product of the marginals, and thus identical.

**Lemma A.4** *The graphs constructed to sample orders for* TSAMPLE *cannot have cycles.*

**Proof**   Let $y_i, y_j, y_k$ form a cycle: $y_i \rightarrow y_j \rightarrow y_k \rightarrow y_i$. By construction, the following conditions must hold for such a cycle to be present:

$$\log p(y_j \mid y_i) - \log p(y_i \mid y_j) > \beta \implies \log p(y_i) < \log p(y_j)$$
$$\log p(y_k \mid y_j) - \log p(y_j \mid y_k) > \beta \implies \log p(y_j) < \log p(y_k)$$
$$\log p(y_i \mid y_k) - \log p(y_k \mid y_i) > \beta \implies \log p(y_k) < \log p(y_i)$$

Putting the three implications together, we get $\log p(y_i) < \log p(y_j) < \log p(y_k) < \log p(y_i)$, which is a contradiction. Hence, the graphs constructed for TSAMPLE cannot have a cycle.

## B   DATASET

| | Input | Output |
|---|---|---|
| Fine-grained emotion classification, [28] (Demszky et al., 2020) | *So there's hope for the rest of us! Thanks for sharing. What helped you get to where you are?* | {curiosity, gratitude, optimism} |
| Open-entity typing [2519] (Choi et al., 2018) | *Some 700,000 cubic meters of caustic sludge and water burst inundating* [SPAN] *three west Hungarian villages* [SPAN] *and spilling.* | {colony, region, location, hamlet, area, village, settlement, community} |
| Reuters [90] (Lewis, 1997) | *India is reported to have bought two white sugar cargoes for. . . . . .cargo sale, they said.* | {ship, sugar} |

Table 3: Real world tasks used for experiments

## C   FIXING THE PROPOSAL DISTRIBUTION IN THE VAE FORMULATION

$$
\begin{aligned}
\log p_\theta(\mathbb{Y} \mid \boldsymbol{x}) &= \log \sum_{\pi_{\boldsymbol{z}} \in \Pi} p_\theta(\pi_z(\mathbb{Y}) \mid \boldsymbol{x}) \\
&= \log \sum_{\pi_{\boldsymbol{z}} \in \Pi} \frac{q_\phi(\pi_{\boldsymbol{z}})}{q_\phi(\pi_{\boldsymbol{z}})} p_\theta(\pi_z(\mathbb{Y}) \mid \boldsymbol{x}) \\
&= \log \mathbb{E}_{q_\phi(\pi_{\boldsymbol{z}})} \left[ \frac{p_\theta(\pi_z(\mathbb{Y}) \mid \boldsymbol{x})}{q_\phi(\pi_{\boldsymbol{z}})} \right] \\
&\geq \mathbb{E}_{q_\phi(\pi_{\boldsymbol{z}})} \left[ \log p_\theta(\mathbb{Y}, \pi_{\boldsymbol{z}} \mid \boldsymbol{x}) \right] - \mathbb{E}_{q_\phi(\pi_{\boldsymbol{z}})} \left[ \log q_\phi(\pi_{\boldsymbol{z}}) \right] \\
\log p_\theta(\mathbb{Y} \mid \boldsymbol{x}) = \log \sum_{\pi_{\boldsymbol{z}} \in \Pi} p_\theta(\pi_z(\mathbb{Y}) \mid \boldsymbol{x}) &\geq \underbrace{\mathbb{E}_{q_\phi(\pi_{\boldsymbol{z}})} \left[ \frac{\log p_\theta(\pi_{\boldsymbol{z}}(\mathbb{Y}) \mid \boldsymbol{x})}{q_\phi(\pi_{\boldsymbol{z}})} \right]}_{\text{ELBO}} = \mathcal{L}(\theta, \phi)
\end{aligned}
$$

$$(7)$$

Where equation 7 is the evidence lower bound (ELBO). The success of this formulation depends on the quality of the proposal distribution $q$ from which the orders are drawn. When $q$ is fixed (e.g., to uniform distribution over the orders), learning only happens for $\theta$. This can be clearly seen from splitting Equation 7 into terms that involve just $\theta$ and $\phi$:

$$\nabla_\phi \mathcal{L}(\theta, \phi) = 0$$
$$\nabla_\theta \mathcal{L}(\theta, \phi) = \nabla_\theta \mathbb{E}_{q_\phi(\pi_{\boldsymbol{z}})} \left[ \log p_\theta(\mathbb{Y}, \pi_{\boldsymbol{z}} \mid \boldsymbol{x}) \right]$$

# D  HYPERPARAMETERS

We list all the hyperparameters in Table 4.

| Hyperparameter | Value |
|---|---|
| GPU | GeForce RTX 2080 Ti |
| gpus | 1 |
| auto_select_gpus | false |
| accumulate_grad_batches | 1 |
| max_epochs | 3 |
| precision | 32 |
| learning_rate | 1e-05 |
| adam_epsilon | 1e-08 |
| num_workers | 16 |
| warmup_prop | 0.1 |
| seeds | [15143, 27122, 999888] |
| add_lr_scheduler | true |
| lr_scheduler | linear |
| max_source_length | 120 |
| max_target_length | 120 |
| val_max_target_length | 120 |
| test_max_target_length | 120 |

Table 4: List of hyperparameters used for all the experiments.

# E  EXPLORING THE INFLUENCE OF ORDER ON SEQ2SEQ MODELS WITH A SIMULATION

We design a simulation to investigate the effects of output order and cardinality on conditional set generation, following prior work that has found simulation to be an effective for studying properties of deep neural networks (Vinyals et al., 2016; Khandelwal et al., 2018).

**Data generation**  We use a graphical model (Figure 3) to generate conditionally dependent pairs $(\boldsymbol{x}, \mathbb{Y})$, with different levels of interdependencies among the labels in $\mathbb{Y}$. Let $\mathbb{Y} = \{y_1, y_2, \ldots, y_N\}$ be the set of output labels. We sample a dataset of the form $\{(\boldsymbol{x}, \boldsymbol{y})\}_{i=1}^m$. $\boldsymbol{x}$ is an $N$ dimensional multinomial sampled from a dirichlet parameterized by $\alpha$, and $\boldsymbol{y}$ is a sequence of symbols with each $y_i \in \mathbb{Y}$. The output sequence $\boldsymbol{y}$ is created in $B$ *blocks*, each block of size $k$. A block is created by first sampling $k - 1$ prefix symbols independently from Multinomial($\boldsymbol{x}$), denoted by $\boldsymbol{y}_p$ The $k^{th}$ suffix symbol ($y_s$) is sampled from either a uniform distribution with a probability $= \epsilon$ or is deterministically determined from the preceding $k - 1$ prefix terms. For block size of 1 ($k = 1$), the output is simply a set of size $B$ sampled from $\boldsymbol{x}$ (i.e., all the elements are independent). Similarly, $k = 2$ simulates a situation with a high degree of dependence: each block is of size 2, with the prefix sampled independently from the input, and the suffix determined deterministically from the prefix. Gradually increasing the block size increases the number of independent elements.

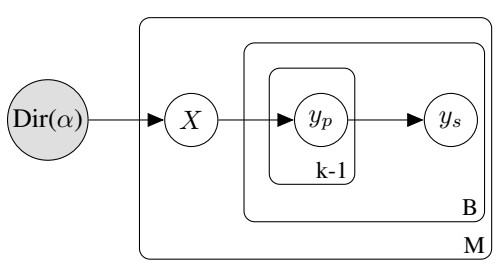

Figure 6: The generative process for simulation

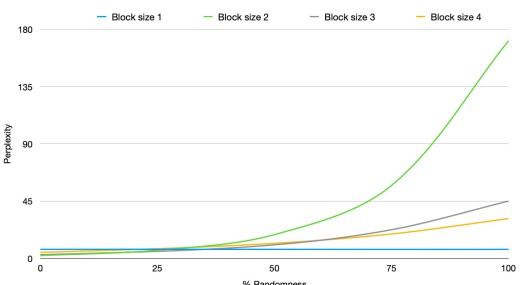

Figure 7: Perplexity vs. Randomness for varying block sizes

### E.1 Major Findings

We now outline our findings from the simulation. We use the architecture of BART-base Lewis et al. (2020) (six-layers of encoder and decoder) without pre-training for all simulations. All the simulations were repeated using three different random seeds, and we report the averages.

**Finding 1: SEQ2SEQ models are sensitive to order, but only if the labels are conditionally dependent on each other.** We train with the prefix $y_p$ listed in the lexicographic order. At test time, the order of is randomized from 0% (same order as training) to 100 (appendixly shuffled). As can be seen from Figure 7 the perplexity gradually increases with the degree of randomness. Further, note that perplexity is an artifact of the model and is independent of the sampling strategy used, showing that order affects learning.

**Finding 2: Training with random orders makes the model less sensitive to order** As Figure 8 shows, augmenting with random order makes the model less sensitive to order. Further, augmenting with random order keeps helping as the perplexity gradually falls, and the drop shows no signs of flattening.

**Finding 3: Effects of position embeddings can be overcome by augmenting with a sufficient number of random samples** Figure 8 shows that while disabling position embedding helps the baseline, similar effects are soon achieved by increasing the random order. This shows that disabling position embeddings can indeed alleviate some concerns about the order. This is crucial for pre-trained models, for which position embeddings cannot be ignored.

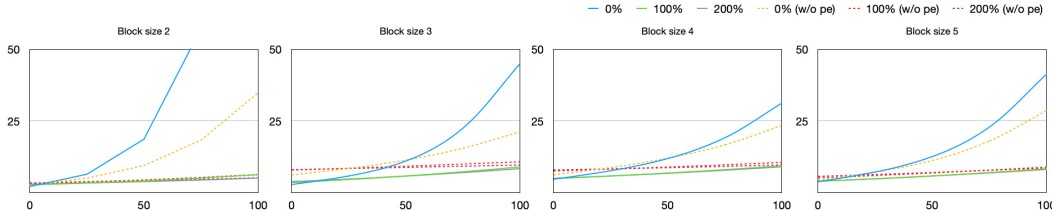

Figure 8: Augmenting dataset with multiple orders help across block sizes. Augmentations also overcome any benefit that is obtained by using position embeddings.

**Finding 4: TSAMPLE leads to higher set overlap** We next consider blocks of order 2 where the prefix symbol $y_p$ is selected randomly as before, but the suffix is set to a special character $y'_p$ with 50% probability. As the special symbol $y'_p$ only occurs with $y_p$, there is a high pmi between each $(y_p, y'_p)$ pair as $p(y_p \mid y'_p) = 1$. Different from finding 1, the output symbols are now shuffled to mimic a realistic setup. We gradually augment the training data with random and topological orders and evaluate the learning and the final set overlap using training perplexity and Jaccard score, respectively. The results are shown in Figure 9. Similar trends hold for larger block sizes, and the results are included in the Appendix in the interest of space.

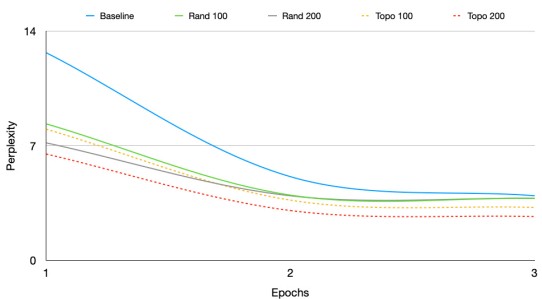 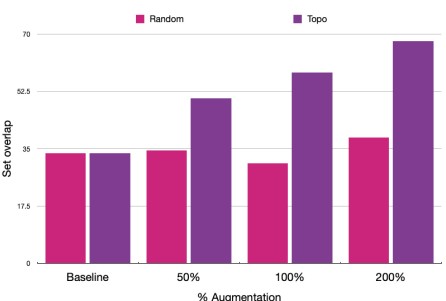

Figure 9: Effect of TSAMPLE on perplexity and set overlap. **Left:** Augmentations done TSAMPLE helps the model converge faster and to a lower perplexity. **Right:** Using TSAMPLE, the overlap between training and test set increases consistently, while consistently outperforming UNIFORM.

|         | Beam           | Random         | Greedy         | Top-k          | Nucleus        |
|---------|----------------|----------------|----------------|----------------|----------------|
| UNIFORM | $0.39 \pm 0.05$ | $0.39 \pm 0.02$ | $0.35 \pm 0.05$ | $0.39 \pm 0.02$ | $0.39 \pm 0.02$ |
| TSAMPLE | $0.67 \pm 0.05$ | $0.67 \pm 0.05$ | $0.71 \pm 0.04$ | $0.67 \pm 0.05$ | $0.68 \pm 0.05$ |

Table 5: Set overlap for different sampling types with 200% augmentations. The gains are consistent across sampling types. Similar trends were observed for 100% augmentation and without positional embeddings. Top-k sampling was introduced by (Fan et al., 2018), and Nucleus sampling by (Holtzman et al., 2020).

**Finding 5: TSAMPLE helps across all sampling types**   We see from Table 5 that our approach is not sensitive to the sampling type used. Across five different sampling types, augmenting with topological orders yields significant gains.

**Finding 6: SEQ2SEQ models can learn cardinality and use it for better decoding**   We created sample data from Figure 6 where the length of the output is determined by sum of the inputs $X$. We experimented with and without including cardinality as the first element. We found that training with cardinality increases step overlap by over 13%, from 40.54 to 46.13. Further, the version with cardinality accurately generated sets which had the same length as the target 70.64% of the times, as opposed to 27.45% for the version without cardinality.

## F   ADDITIONAL RESULTS

We present all the results for the three tasks in Tables 6, 7, and 8.

### F.1   CLASSIFICATION RESULTS WITH BERT

Table 9 includes results from a multi-label classification baseline where bert-base-uncased is used as the encoder.

## G   SAMPLE GRAPHS

In this section, we present examples from REUTERS and GO-EMO datasets to further understand the permutations generated by our method.

**What kinds of permutations does TSAMPLE create?**   As discussed in Section 3.1, TSAMPLE encourages highly co-occuring pairs $(y_i, y_j)$ to be in the order $y_i, y_j$ if $p(y_j \mid y_i) > p(y_i \mid y_j)$. In our analysis, this dependency in the datasets shows that the orders exhibit a pattern where *specific* labels appear before the *generic* ones. For example, in case of entity typing, the more generic entity *event* is generated after the more specific entities *home game* and *match* Figure 4.3 (left).

| | $p_{\text{micro}}$ | $p_{\text{macro}}$ | $r_{\text{micro}}$ | $r_{\text{macro}}$ | $F_{\text{micro}}$ | $F_{\text{macro}}$ | $jaccard$ |
|---|---|---|---|---|---|---|---|
| SET SEARCH | 47.17 | 10.68 | 13.09 | 7.02 | 10.7 | 7.36 | 7.4 |
| SEQ2SEQ | 41.65 | 27.39 | 35.19 | 26.21 | 27.4 | 23.41 | 23.4 |
| SEQ2SEQ + CARD | 39.77 | 33 | 38.02 | 28.31 | 33 | 26.79 | 26.8 |
| UNIFORM + CARD | 44.77 | 35.6 | 32.96 | 26.54 | 35.6 | 27.53 | 27.5 |
| TSAMPLE + CARD | 43.37 | 36.08 | 34.51 | 30.54 | 36.1 | 30.01 | 30 |
| UNIFORM- CARD | 48.85 | 32.45 | 27.75 | 19.86 | 32.5 | 22.67 | 22.7 |
| TSAMPLE- CARD | 50 | 36.68 | 29.84 | 19.84 | 36.7 | 23.31 | 23.3 |

Table 6: Results for GO-EMO.

| | $p_{\text{micro}}$ | $p_{\text{macro}}$ | $r_{\text{micro}}$ | $r_{\text{macro}}$ | $F_{\text{micro}}$ | $F_{\text{macro}}$ | $jaccard$ |
|---|---|---|---|---|---|---|---|
| SET SEARCH | 70.04 | 10.92 | 34.9 | 7.1 | 46.56 | 7.54 | 37.49 |
| SEQ2SEQ | 66.36 | 24.74 | 42.28 | 13.78 | 51.64 | 15.58 | 44.3 |
| SEQ2SEQ + CARD | 73.02 | 34.17 | 53.8 | 21.85 | 61.95 | 24.28 | 59.08 |
| UNIFORM + CARD | 74.26 | 35.31 | 54.33 | 22.13 | 62.75 | 24.74 | 58.95 |
| TSAMPLE + CARD | 75.65 | 36.67 | 55.54 | 24.13 | 64.05 | 26.66 | 61.14 |
| UNIFORM- CARD | 69.56 | 26.68 | 38.15 | 12.71 | 49.27 | 15.2 | 42.24 |
| TSAMPLE- CARD | 76.55 | 26.49 | 41.78 | 12.77 | 54.06 | 15.78 | 47.34 |

Table 7: Results for REUTERS.

| | $p_{\text{micro}}$ | $p_{\text{macro}}$ | $r_{\text{micro}}$ | $r_{\text{macro}}$ | $F_{\text{micro}}$ | $F_{\text{macro}}$ | $jaccard$ |
|---|---|---|---|---|---|---|---|
| SET SEARCH | 24.65 | 26.5 | 29.98 | 31.44 | 23.92 | 26.25 | 13.39 |
| SEQ2SEQ | 52.78 | 55.4 | 39.84 | 42.42 | 41.45 | 44.63 | 24.6 |
| SEQ2SEQ + CARD | 61.26 | 62.48 | 41.87 | 44.68 | 48.07 | 50.48 | 27.84 |
| UNIFORM + CARD | 67.56 | 68.59 | 39.61 | 42.25 | 47.98 | 50.4 | 26.89 |
| TSAMPLE + CARD | 64.58 | 65.53 | 44.6 | 47.46 | 51.2 | 53.48 | 29.39 |
| UNIFORM- CARD | 60.93 | 62.57 | 39.09 | 41.69 | 44.2 | 46.85 | 25.26 |
| TSAMPLE- CARD | 58.02 | 59.88 | 42.63 | 44.95 | 46.54 | 48.86 | 26.82 |

Table 8: Results for OPENENT.

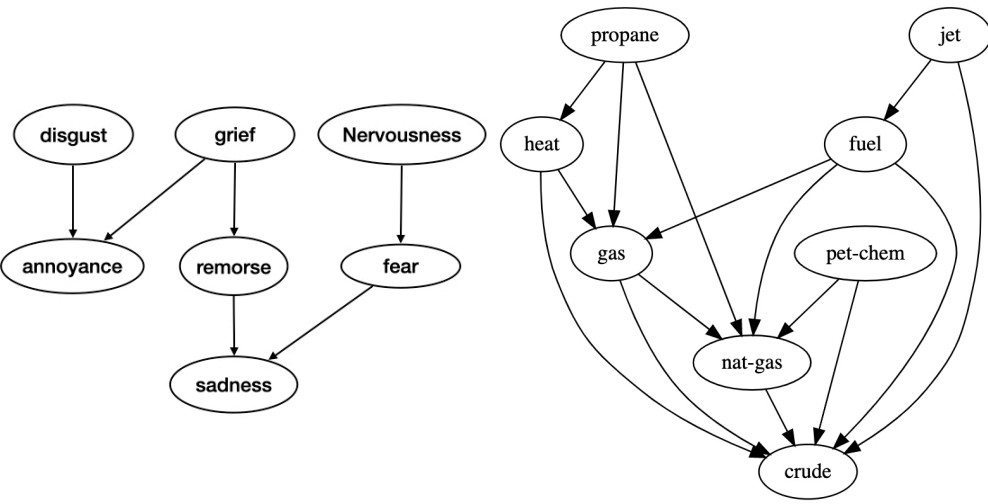

Figure 10: Label dependencies used by TSAMPLE for GO-EMO (left) and REUTERS (right) shows that the method puts specific entities before generic ones.

| | GO-EMO | | | OPENENT | | | REUTERS | | | |
|---|---|---|---|---|---|---|---|---|---|---|
| | $p$ | $r$ | $F$ | $p$ | $r$ | $F$ | $p$ | $r$ | $F$ | |
| BERT @1 | 31.8 | 10.3 | 15.6 | 38.0 | 10.3 | 15.9 | 31.7 | 12.3 | 17.6 | |
| BERT @3 | 23.8 | 23.4 | 23.6 | 19.7 | 14.0 | 16.1 | 23.4 | 28.3 | 25.5 | |
| BERT @5 | 20.6 | 34.0 | 25.7 | 15.5 | 18.0 | 16.4 | 18.8 | 37.6 | 24.9 | |
| BERT @10 | 16.5 | 54.3 | 25.3 | 11.8 | 26.0 | 16.0 | 15.1 | 61.8 | 24.2 | |
| BERT @20 | 14.1 | 93.2 | 24.5 | 8.4 | 34.3 | 13.5 | 9.5 | 75.9 | 16.8 | |
| BERT @50 | - | - | - | 2.6 | **50.2** | 4.9 | 8.9 | - | - | - |
| BERT | 21.4 | 43.0 | 22.9 | 16.0 | 25.5 | 13.8 | 19.7 | 43.2 | 21.8 | |
| BART @1 | 31.7 | 10.3 | 15.5 | 38.0 | 10.3 | 15.6 | 31.8 | 12.3 | 17.6 | |
| BART @3 | 21.2 | 21.0 | 21.0 | 19.7 | 14.0 | 15.8 | 23.1 | 28.1 | 25.2 | |
| BART @5 | 14.1 | 33.4 | 25.6 | 15.5 | 18.0 | 16.2 | 18.7 | 37.6 | 24.8 | |
| BART @10 | 16.3 | 53.4 | 25.0 | 11.7 | 26.0 | 15.9 | 15.1 | 62.0 | 24.1 | |
| BART @20 | 14.1 | **93.3** | 24.5 | 8.4 | 34.3 | 13.4 | 9.6 | **77.1** | 17.1 | |
| BART @50 | - | - | - | 4.9 | 48.0 | 8.9 | - | - | - | |
| BART | 20.8 | 42.4 | 22.4 | 16.4 | 25.1 | 14.3 | 19.7 | 43.4 | 21.7 | |
| SET SEARCH | 10.7 | 7.0 | 7.4 | 26.5 | 31.4 | 26.3 | 10.9 | 7.1 | 7.5 | |
| SEQ2SEQ | 27.4 | 26.2 | 23.4 | 55.4 | 42.4 | 44.6 | 24.8 | 13.8 | 15.6 | |
| UNIFORM | 32.5 | 19.9 | 22.7 | 62.6 | 41.7 | 46.9 | 26.7 | 12.7 | 15.2 | |
| TSAMPLE | **36.7** | 19.8 | 23.3 | 60.0 | 44.5 | 48.0 | 26.5 | 12.8 | 15.8 | |
| SEQ2SEQ +CARD | 33.0 | 28.3 | 26.8 | 62.5 | 44.7 | 50.5 | 34.1 | 21.8 | 24.3 | |
| UNIFORM + CARD | 35.6 | 26.5 | 27.5 | **68.6** | 42.3 | 50.4 | 35.3 | 22.1 | 24.7 | |
| TSAMPLE + CARD | 36.1 | **30.5** | **30.0** | 65.5 | **47.5** | **53.5** | **36.7** | **24.1** | **26.7** | |

Table 9: Our main results: using permutations generated by TSAMPLE and adding cardinality gives the best overall performance in terms of macro precision, recall, and $F$−score score. Statistically significant results are underscored. CARD stands for cardinality. BERT @k / BART @k denotes the pointwise classification baseline using BERT/ BART where the top $k$ labels are used as the model output. The average is denoted by BERT/ BART.

