# OpenReview forum: "Conditional set generation using Seq2seq models"
_ICLR.cc/2022/Conference — ICLR 2022 Submitted_

### Official Review · Reviewer_MDhc · 2021-10-31

**Correctness:** 3
**Technical Novelty And Significance:** 2
**Empirical Novelty And Significance:** 2
**Recommendation:** 5
**Confidence:** 3

**Main Review:**

Strength:
- The introduction of the order as the latent variables is reasonably intuitive and well-motivated.
- The proposed approach of jointly modeling the cardinality with topological sort-based TSAMPLE data augmentation seems interesting.
- The authors promise the availability of corresponding code and provide a detailed description of hyperparameters which would be essential for reproducibility if open-sourced.

Weakness:

- Missing state-of-the-art comparisons. Relevant baselines have not been used for this task, such as non-seq2seq or classification-based models.
- It is not clear how this approach would generalize in a zero-shot setting where the partial order graph is not known beforehand.
- An ablation study is required to understand how the model is performing for different set lengths in Section 4.3 (Role of cardinality). It is expected that the model might be performing well for shorter set lengths, which could help understand the impact of the proposed approach.
- Some model-generated examples would provide more insight.
- Human evaluation is not provided.
- Pictorial depiction of the models either in the main section or in the appendices would increase the understanding and interpretability of the proposed approach.

Questions:
- On page 7, could the authors specify the type of sampling (greedy/beam/etc.) used to report results in the rest of the main text (in table 2 etc.)?
- Have the authors experimented with pre-trained weights (decoders)?
- Section 5 Conclusion -> Could the authors provide more information as to how they could include cardinality in dialog generation and how would it help?

Suggestions/Comments:
- Section 1 can be improved significantly by providing more concrete real-life examples or discussing Appendix B (Table 3).
- Please take care of using citep compared to citet (natbib style) appropriately. Eg on Page 6.
- Please provide a more detailed description of |sYt| in Section 3.2
- Section 4.2 Model networkx -> network
- performance SET SEARCH -> performance of SET SEARCH
- Figure 5 Right -> It would help to also include (T) in the legend to indicate TSAMPLE
- Section 4.3 Role of cardinality: results 2 -> Table 2
- It would help to differentiate between the definition of `n’ in Section 4.1 and Section 4.2 baselines and in the main text.
- Some missing references:
https://aclanthology.org/P19-1139.pdf
https://aclanthology.org/2021.findings-acl.121.pdf


**Summary Of The Paper:**

This paper explores the task of conditional set generation using sequence generation models. The authors propose to model order-invariance and set cardinality into the seq2seq models.
Additionally, the authors introduce a novel data augmentation approach based on topological sorting. The proposed model is shown to improve performance when compared to seq2seq baselines on three different datasets.


**Summary Of The Review:**

The results seem promising; however, the paper could be improved in the presentation and more detailed descriptions with comparisons to the SOTA models.

---

> ### Author Response · Authors · 2021-11-21
> **Response to reviewer MDhc**
>
> Thanks for your feedback.
>
> > Missing state-of-the-art comparisons
>
> - We have addressed this in the main rebuttal. Specifically, we have compared our methods with BART and BERT-based standard multi-label models.
>
> > It is not clear how this approach would generalize in a zero-shot setting where the partial order graph is not known beforehand.
>
> - This is an exciting future direction. Unfortunately, the zero-shot generalizability of our approach is beyond the scope of the current paper, and we plan to explore this in future work.
>
> > An ablation study is required to understand how the model is performing for different set lengths in Section 4.3 (Role of cardinality).
>
> - We conducted an ablation study by evaluating the performance of our method for different set lengths. We find that the impact of cardinality on performance is dataset-dependent. For example, while the F1 does go down with cardinality in the case of Reuters, it increases before plateauing for OpenEnt. The plots are available at the following anonymized URLs: [Reuters](https://i.ibb.co/y0GXmgL/F1-vs-number-of-elements-Reuters.png), [OpenEnt](https://i.ibb.co/12rcCFZ/F1-vs-number-of-elements-Open-ENT.png)
>
> > Some model-generated examples would provide more insight.
>
> - We have added additional examples in Appendix G of the updated draft.
>
> > Human evaluation.
>
> - Thanks for bringing this up. We conduct a human evaluation to compare the outputs generated by the vanilla seq2seq model with our best approach. Since automated metrics already capture many aspects of the performance of the models, we focus on a common pitfall of generation models: the generation of ill-formed output. For the open-entity typing dataset, we sample 100 random examples and check whether the generated set contains any element that is ill-formed or out-of-vocabulary. For the 100 examples, we find that the vanilla seq2seq approach generates sets with an ill-formed element 22% of the time, whereas our best model completely avoids this. Examples of such ill-formed elements include _personformer, businessirm, polit, foundationirm, politplomat, eventlete_. Crucially, note that both the models were trained for the same number of steps. This analysis indicates that training the model with an informative order infuses more information about the underlying type-hierarchy, avoiding the ill-formed elements.
>
> > Pictorial depiction of the models either in the main section or in the appendices would increase the understanding and interpretability of the proposed approach.
>
> - With more space available to us in the camera-ready, we would include examples from Table 5 and Figure 10 (currently in the Appendix) to the main paper.
>
> > On page 7, could the authors specify the type of sampling (greedy/beam/etc.) used to report results in the rest of the main text (in table 2 etc.)?
>
> - We use greedy decoding. In Table 5, we show results on simulations with five different samplings and find that the sampling method does not affect the performance.
>
> > Have the authors experimented with pre-trained weights (decoders)?
>
> - Yes. All the models (including baseline and the experiments reported in this response) use pretrained weights (for either bart-base or bert-base-uncased). Please note that the architecture of the bert-base-uncased is different from bart~(encoder only).
>
> > Section 5 Conclusion -> Could the authors provide more information as to how they could include cardinality in dialog generation and how would it help?
>
> - For open-ended dialog generation, we plan to first identify instances in dialog where cardinality is essential. For example, queries such as “Can I get 5 exercises to do at home ?” - We anticipate that incorporating our existing approach to questions that require cardinality might help improve machine responses to such queries. Further, anticipating the length of the expected response might help the dialog system utilize the right knowledge source for generating the response. For instance, if the expected response is longer than a few sentences, the system might use external knowledge sources to generate a helpful answer.
>
>
> > sY
>
> - Sorry, that was a typo (should have been \sY) that we have fixed. We have also included your other comments in the main text. Thanks for your inputs.

---

### Official Review · Reviewer_NrsZ · 2021-11-02

**Correctness:** 3
**Technical Novelty And Significance:** 2
**Empirical Novelty And Significance:** 3
**Recommendation:** 5
**Confidence:** 4

**Main Review:**

I appreciate the detail that the authors have given to describing the imposition of order on labels using PMI, and find it interesting that the pairwise order threshold beta does not seem to impact the results.

I would like the authors to expand on the storage complexity (|Y|^2 in the worse case) and how it could impact cases where this model may be applied to dense multilabel data.

I would like to see a comparison between TSAMPLE and other seq2seq set generation baselines and non-seq2seq methods (e.g. a multi-label classification and/or pairwise scoring approach e.g. [2, 3]) to assess whether there is a large gap and thus whether there is a fundamental difference in applicability of a seq2seq approach to set generation based on the task and label space. While the authors claim that their method is particularly useful for tasks such as OpenEnt, the relatively small label vocabulary (2.5K) makes ranking/scoring and classification (extreme or otherwise) based approaches reasonable to apply. The unique label space is even smaller in GO-EMO and REUTERS.

Did the authors compare their method with naive usage of the seq2seq model (BART-base) without set constraints at training time? This would represent the most naive baseline that may still perform relatively well, given the inclusion of a simple post-processing step that removes repeated predictions in a sequence. This post-processing step seems necessary regardless, as the proposed method here does not explicitly constrain the generated sequence to contain unique elements.

I would like to see a comparison with previous work that constrains seq2seq models directly to generate sets without data augmentation, such as [1] which incorporates a cardinality penalty and changes the decoding strategy for a seq2seq model to generate sets. This accommodates the same base model architecture, and has been applied to a set prediction task (Ingredient prediction) at a scale that is comparable to the larger of datasets in this paper (e.g. OpenEnt).

Small edit: Are the max/min labels in Table 1 reversed?

References:
[1] Salvador, Amaia, et al. "Inverse cooking: Recipe generation from food images." Proceedings of the IEEE/CVF Conference on Computer Vision and Pattern Recognition. 2019.
[2] Dai, Hongliang et al. “Ultra-Fine Entity Typing with Weak Supervision from a Masked Language Model.” ACL/IJCNLP (2021).
[3] Wu, Ledell, et al. "Scalable Zero-shot Entity Linking with Dense Entity Retrieval." Proceedings of the 2020 Conference on Empirical Methods in Natural Language Processing (EMNLP). 2020.

**Summary Of The Paper:**

The authors propose to perform set (order-invariant) generation with seq2seq models via two concepts: 1) to impose an informative order over labels using a fixed graph ordering, and 2) explicitly predicting the cardinality (size) of the predicted set. The authors ablate their approach against other data augmentation / training data ordering methods on three different datasets/tasks, showing that TSAMPLE results in significant F1 score improvements compared to these ablations.

**Summary Of The Review:**

This work provides an interesting data augmentation approach to training set generation models using seq2seq formulations. I would like to see comparisons against more fair baselines as well as non-seq2seq approaches to better contextualize the applicability of this work, especially with regards to methods for seq2seq set generation that do not rely on data augmentation.

---

> ### Author Response · Authors · 2021-11-21
> **Response to reviewer NrsZ**
>
> Thanks for your feedback.
>
> > I would like the authors to expand on the storage complexity (|Y|^2 in the worse case) and how it could impact cases where this model may be applied to dense multi-label data.
>
> - Indeed, our method will not scale in settings where |Y| is large and most combinations of labels appear together. However, for large |Y| (e.g., |Y| ~ 10^6), as the number of |Y|^2 combinations that appear in the data increases, the dataset size will proportionally grow to intractable lengths (in this case, there will be 10^12 points in the dataset). Thus, we don't expect this to be an issue in practice.
> - We also note that our method easily scaled for the keyphrase generation dataset, despite a large label space (250k+ labels), providing additional support to the hypothesis that a large label space might not be an issue for our method in practice.
>
> > Other baselines, large label-space
>
> - We have added other baselines in the common response, including one that uses a large label-space of 250k labels. We hope that these additional results address the concerns.
>
> > Did the authors compare their method with naive usage of the seq2seq model (BART-base) without set constraints at training time?
>
> - We apologize for the lack of clarity; the Seq2Seq baseline in the paper is essentially the naive usage of the seq2seq model for the task (the cardinality is not predicted, and we do not perform any data augmentation).
>
> > I would like to see a comparison with previous work that constrains seq2seq models directly to generate sets without data augmentation
>
> - We hope that the experiments added in the common response address this concern. Further, please note that the Seq2seq baseline does not use any data augmentation method~(but trains on the exact same amount of data, to rule out the possibility of improvements being only due to longer training times).
>
> > The min/max labels
>
> - Thanks for pointing this out; the labels were reversed. We have cited the missing references in the updated submission appropriately.

---

### Official Review · Reviewer_vizW · 2021-11-02

**Correctness:** 3
**Technical Novelty And Significance:** 3
**Empirical Novelty And Significance:** 3
**Recommendation:** 6
**Confidence:** 4

**Main Review:**

Strengths:
- The paper proposes a simple, interesting idea and deals with a timely topic.
- The experiments are described thoroughly and seem well-executed.

Weaknesses:
- My main concern is that seq2seq is unnecessary for tackling the kinds of problems the authors consider, namely, where all target sets are subsets of a finite set of discrete elements. Here it seems more natural to simply make a binary prediction for each set element. Note that such an approach is compatible with first predicting cardinality and also with using pretrained components (like BART), at least on the encoder side. It's possible that seq2seq would outperform such an approach, but the authors do not provide evidence of this.
- More minor, since it doesn't affect the rest of the paper much, but Lemma A.1 and its proof seem incorrect.

Update after author response: Thanks for your response and your comments. I'm increasing my score in response to clarifications and the improved results over baseline multi-label classifiers.

**Summary Of The Paper:**

The paper proposes an approach to set-generation within a seq2seq framework. The authors propose to train a standard seq2seq model by ordering the discrete elements in the target sets (as a sequence) under a partial order defined by taking $y_i < y_j$ if both $y_i$ and $y_j$ have sufficiently high PMI and $p(y_i | y_j)$ is sufficiently larger than $p(y_j | y_i)$. In particular, the authors sample from orderings consistent with this partial order during training. The authors show that this approach improves over baseline seq2seq approaches for set generation on synthetic and NLP tasks, and also that prepending the cardinality of the set to be predicted to the target sequence is generally helpful.

**Summary Of The Review:**

The paper proposes a simple, interesting idea, but needs additional baselines to show the proposed approach is useful.

---

> ### Author Response · Authors · 2021-11-21
> **Response to reviewer vizW**
>
> Thanks for your feedback.
>
> > My main concern is that seq2seq is unnecessary for tackling the kinds of problems the authors consider, namely, where all target sets are subsets of a finite set of discrete elements.
>
> - This is a valid concern. However, for NLP tasks, we would like to point some evidence and justification towards our approach by arguing the following:
>
> 1. It might be computationally infeasible to perform standard multi-label classification for a dataset with a large number of labels. To verify this, we add an additional multi-label classification task (keyphrase generation) with ~250k labels in the common response. While seq2seq models can be readily adapted for this task, it is not tractable to train a standard multi-label classifier (i.e., making a binary prediction for each label) for a label-set this large.
>
> 2. Additionally, in NLP tasks, pretrained seq2seq models gain leverage from learning during pretraining. Such methods have recently shown state-of-the-art performance on tasks like summarization that usually have sequences that are > 200 tokens. We believe that our approach provides a way of leveraging pre-trained language models for predicting a finite set of discrete elements in NLP settings. Further, our empirical results show that our method outperforms the standard multi-label classification baselines.
>
> > Note that such an approach is compatible with first predicting cardinality and also with using pretrained components (like BART), at least on the encoder side. It's possible that seq2seq would outperform such an approach, but the authors do not provide evidence of this.
>
> - We have conducted experiments with both these approaches (we use bert as an encoder only baseline) and have reported the results in the updated draft and the common response. Seq2seq models do outperform, as you have hinted.
> - Regarding cardinality, we perform inference with varying values of $k$, and find that our method improves over the baseline for all values of $k$. This indicates that even if the optimal cardinality was known, the point-wise prediction baseline would underperform our approach.
>
> > Lemma A.1
>
> - We have added an additional condition that the distributions have to be non-negative. We hope this clarifies the issue. For camera-ready, we will scrutinize the proofs further and make appropriate changes if needed.

---

### Official Review · Reviewer_ue1r · 2021-11-03

**Correctness:** 3
**Technical Novelty And Significance:** 2
**Empirical Novelty And Significance:** 2
**Recommendation:** 5
**Confidence:** 4

**Main Review:**

Strengths:
- The motivation that using order and cardinality for set generation task is clearly expressed.
- Using topology sort to generated sorted label is an attracting idea.

Weaknesses:
- The F1 score of GO-EMO in the original paper (Demszky et al. 2020) is 46 (using BERT-base). However, in Table 3, the best score is only 30. Can you explain the difference here?
- Stronger baselines are missing in Table 2. Although sorting the labels is important, the effectiveness of using mutual information for ordering is unknown. It would be better to compare with other sorting methods.
- When using partial order to construct the graph, it is possible that there will be circles. It’s not clear how this case is handled.
- As the cardinality is predicted, can we use it to control the label generation progress? E.g., use it to decide when to stop generation instead of <EOS>.
- Typos:
      o  The recent successes of pretraining-finetuning paradigm has => The recent successes of pretraining-finetuning paradigm have
      o  From the results 2 => From the results in Table 2


**Summary Of The Paper:**

In this paper, the authors proposed a data augmentation approach to improve the conditional set generation task. The topology sort is used to recovers the informative orders of labels, and set cardinality is added to the target sequence. The authors conducted experiments on simulated data and three NLP data sets to demonstrate the effectiveness of the method.

**Summary Of The Review:**

This authors propose to use PMI for label sorting in the set generation. However, the experiment only demonstrates the importance of sorted labels is better than random permutation, but not how effective it is when compared with other sorting methods.

---

> ### Author Response · Authors · 2021-11-21
> **Response to reviewer ue1r**
>
> Thanks for your feedback.
>
> > The F1 score of GO-EMO in the original paper (Demszky et al. 2020) is 46 (using BERT-base). However, in Table 3, the best score is only 30. Can you explain the difference here?
>
> - We use different splits (we *only* retain samples with more than one label, section 4.2). To address this concern, we also report our experimental numbers on the full splits of the dataset (below).
>
> |                | P     | R     | F     |
> |----------------|-------|-------|-------|
> | Demszky et al. | 51.74 | 44.23 | 47.55 |
> | our method     | 61.48 | 51.58 | 53.59 |
>
> - On our split (datapoints with >1 labels), the implementation proposed by Demszky et al. 2020 achieves a macro F1 of 26 using BERT-base, and our approach achieves 30 F1.
>
> - *Key Finding*:  Our method outperforms Demsezky et al. 2020 by a considerable margin.
>
> > Stronger baselines are missing in Table 2.
>
> - We hope that the common response and the experiments vs. Demsezky et al. 2020 are useful in addressing the concerns regarding comparison with strong baselines.
>
> > Although sorting the labels is important, the effectiveness of using mutual information for ordering is unknown. It would be better to compare with other sorting methods.
>
> - We want to clarify that the baselines (seq2seq, random) do not use any sorting mechanism for ordering labels (i.e., the labels are randomly listed). In contrast, our approach uses PMI & conditional probability criteria for sorting the labels. Thus, the primary difference between these baselines and our method is the sorting method. A good extension of our current approach is to find other sorting mechanisms, and we plan to explore them in the future.
>
> - To check this further, we perform data augmentation using the reverse of the order given by our method, and we find that such a reversed order yields a drop of 12\% in the F1 score. Please see Section 4.3, "Reversing the order," for more details. Please let us know if we misunderstood the question.
>
> > On cycles
>
> - Our method avoids cycles by construction, and we provide a proof in the (updated) Appendix A.4. In all of our four datasets, we did not encounter any cases with cycles. To avoid any practical errors (e.g., corner cases), we ignore the cycles in our implementation.
>
> > As the cardinality is predicted, can we use it to control the label generation progress? E.g., use it to decide when to stop generation instead of <EOS>.
>
> - We find that cardinality is implicitly being used for deciding <EOS> generation. For all datasets, we find that number of generated labels agrees with actual cardinality in over 90% of cases, with a margin of $\pm 1$.
>
> ---

---

### Author Response · Authors · 2021-11-21
**Common response to all reviewers**

We thank all the reviewers for their insightful comments and feedback. We are encouraged that the reviewers find our approach well-motivated and interesting. We add the following experiments for addressing the common concerns regarding non-seq2seq baselines, and to show additional evidence for the utility of our approach:

1. For all the datasets, we run the multi-label classification baseline using BART-Base (same as our base model) and BERT-Base. This experiment establishes a strong non-seq2seq baseline for our approach.

2. We also added another dataset: KP-20k~([1]). This dataset is uniquely suited to compare our approach to strong baselines since the label-space in this dataset is large (257k labels), and standard multi-label classification baselines are not computationally feasible.

---

## Comparison with non-seq2seq baselines

We perform experiments with two models:

1. BART-Base (direct comparison to our model setting since our model is based on BART-Base): In this setting, we adapt the BART-Base model to perform multi-label classification by adding a classification layer to BART-Base Encoder.

2. BERT-Base: In this setting, we use bert-base-uncased as the encoder, which is fed to a classification head. Results for this model are shown in the updated Appendix (Section F.1).

We report results from the three experiments (averaged over three random seeds) using bart-base. Our results using both BART-Base and BERT-Base show similar trends, and we found that our method performs better than both these baselines. Following the standard multi-label setup, we use a hyperparameter $k$ for classification where we pick top-$k$ labels as our predictions.

### Key Findings

1. Our approach outperforms the multi-label classification approach (results below).

2. Additionally, we perform inference with different values of $k$ (please note that choosing $k$ is not required with our approach) and find that our method outperforms the baseline in all the settings.

| | GO-EMO | | |   OPENENT  |        |        |       REUTERS       |        |        |
|---------------|:---------:|--------------:|--------------:|:-----------:|---------------------:|---------------------:|:--------------------:|---------------------:|---------------------:|
| |    $p$   |      $r$     |      $F$     |     $p$    |         $r$         |         $F$         |         $p$         |         $r$         |         $F$         |
| bart-multi-label@1       | 31.7      | 10.3 | 15.5 |        38.0 |   10.3 |   15.6 |   31.8 |   12.3 |   17.6 |
| bart-multi-label@3       | 21.2      | 21.0 | 21.0 |        19.7 |   14.0 |   15.8 |   23.1 |   28.1 |   25.2 |
| bart-multi-label@5       | 14.1      | 33.4 | 25.6 |        15.5 |   18.0 |   16.2 |   18.7 |   37.6 |   24.8 |
| bart-multi-label@10      | 16.3      | 53.4 | 25.0 |        11.7 |   26.0 |   15.9 |   15.1 |   62.0 |   24.1 |
| bart-multi-label@20      | 14.1      | **93.3**  | 24.5 |         8.4 |   34.3 |   13.4 |    9.6 |       **77.1** |   17.1 |
| bart-multi-label@50      | -         |    - |    - |         4.9 |        **48.0** |    8.9 |      - |      - |      - |
| bart-multi-label (average)        | 20.8      | 42.4 | 22.4 |        16.4 |   25.1 |   14.3 |   19.7 |  43.4 |   21.7 |
| TSAMPLE + CARD | 36.1      | 30.5 | **30.0** | 65.5 | **47.5** | **53.5** | **36.7** | 24.1 | **26.7** |

---

## Additional experiments on KP-20k classification dataset

- To further motivate our method (i.e., the utility of seq2seq models for set prediction tasks), we experiment on KP-20k, which is an extreme multi-label classification dataset ([1]). In this dataset, the input text is an abstract from a scientific paper, and the output is a set of keyphrases~(label space is 257k). We perform the "absent keyphrase generation" task in this experiment and follow the same experimental setup as the other datasets.

 The data statistics are as follows:

- Train/test/dev points: 157k/2k/2k.
- Labels: Total: 257k, 3.87 labels/ sample average.

### Key Findings on KP-20k

- We find that our approach outperforms the baseline in the KP-20k dataset.
- This setup also highlights the utility of a seq2seq approach for the set generation task: a standard multi-label classification cannot be easily scaled in this setting, whereas adapting a seq2seq model is straightforward.

|    |$p$   |      $r$     |      $F$   |
|-------------------- |------|------|------|
| Ye et al. 2021      | -    |   -  | 5.80 |
| Seq2seq             | 6.67 | 5.53 | 5.90 |
| Seq2seq + CARD      | 7.06 | 5.65 | 6.16 |
| UNIFORM + CARD      | 7.32 | 5.75 | 6.30 |
| TSAMPLE + CARD (Our method)     | **7.70** | **6.06** | **6.64** |

[1] Meng, Rui, Sanqiang Zhao, Shuguang Han, Daqing He, Peter Brusilovsky, and Yu Chi. "Deep keyphrase generation." arXiv preprint arXiv:1704.06879 (2017).

[2] Ye, J., Gui, T., Luo, Y., Xu, Y., & Zhang, Q. (2021). ONE2SET: Generating Diverse Keyphrases as a Set. arXiv preprint arXiv:2105.11134.

---

### Decision · Program_Chairs · 2022-01-20

**Decision:**

Reject

**Comment:**

In this paper, the authors propose a method to generate sets, which are order invariant, with a sequence-to-sequence model. The main idea is to order the elements of the sets, and then treat them as regular sequences. The authors propose to use PMI and conditional probability to obtain a partial order on the elements of sets. Overall, while the reviewers note that the proposed method is simple and intuitive, they also raised concerns about the paper: one of the main concerns is about missing baselines, such as non seq2seq models for set generation, such as binary classification (to predict whether an element should be included or not). For this reason, I recommend to reject the paper.